# “Life Is Taking Me Where I Need to Go”: Biographical Disruption and New Arrangements in the Lives of Female Family Carers of Children with Congenital Zika Syndrome in Pernambuco, Brazil

**DOI:** 10.3390/v12121410

**Published:** 2020-12-08

**Authors:** Ana Paula Lopes de Melo, Tereza Lyra, Thália Velho Barreto de Araújo, Maria do Socorro Veloso de Albuquerque, Sandra Valongueiro, Hannah Kuper, Loveday Penn-Kekana

**Affiliations:** 1Collective Health Centre, Academic Centre of Vitoria, Federal University of Pernambuco, Vitoria de Santo Antão 55608-680, Brazil; 2Aggeu Magalhães Institute, Oswaldo Cruz Foundation, Pernambuco 50670-420, Brazil; tmlyra@gmail.com; 3Department of Social Medicine, Faculty of Medicine, University of Pernambuco, Recife 50670-901, Brazil; 4Postgraduate Programme in Public Health, Centre of Medical Sciences, Federal University of Pernambuco, Recife 50670-901, Brazil; thalia.velhobarreto@gmail.com (T.V.B.d.A.); msoveloso@gmail.com (M.d.S.V.d.A.); svalong@gmail.com (S.V.); 5International Centre for Evidence in Disability, Clinical Research Department, London School of Hygiene and Tropical Medicine, London WC1E 7HT, UK; hannah.Kuper@lshtm.ac.uk; 6Department of Infectious Disease Epidemiology, London School of Hygiene and Tropical Medicine, London WC1E 7HT, UK; loveday.Penn-Kekana@lshtm.ac.uk

**Keywords:** congenital Zika syndrome, maternal health, gender, family caregivers, biographical disruption

## Abstract

The congenital Zika syndrome (CZS) epidemic in Brazil turned the spotlight on many other factors beyond illness, such as poverty, gender, and inequalities in health care. Women were the emblematic subjects in this study, not only because Zika virus is a vertical transmission disease, but also because women—in Brazil and elsewhere—typically represent the primary carers of children. This is a qualitative analytic study using semi-structured interviews with 23 female family carers of children with CZS in Brazil. Through the concept of biographical disruption, we analysed some of the social impacts experienced by women involved in caring for affected children. We identified that the arrival of a child with disabilities resulted in biographical disruption similar to that experienced by people with chronic illnesses. Social support networks were configured through an alliance between women from different generations, revealing solidarity networks, but also highlighting the absence of the state in tackling these social vulnerabilities. Tracing the pathways of these biographical narratives enables us to understand how women have acted to defend the value of their disabled children in a society structured on the model of body normativity and inequality. These results may provide clues to a more inclusive society, which confronts systems of gender oppression and the sexual division of labour focused on women.

## 1. Introduction

Zika virus infection had its greatest global impact between 2015 and 2016 when a large-scale outbreak of microcephaly in newborns in Brazil began to be associated with Zika infection during pregnancy [1]. The World Health Organization declared an international emergency for almost the whole of 2016, considering this virus as a new global threat because of its significant epidemic potential [2]. Since then, approximately 80 countries have reported outbreaks of Zika, and approximately 30 have declared cases of congenital Zika syndrome (CZS) [3]. A review conducted by Pomar et al. demonstrated that the maternal–fetal transmission rate remains difficult to estimate, but some studies point to a range between 7% and 26%, depending on the investigation methods applied. Although the definition of CZS remains to be established, a wide spectrum of adverse effects have been described in relation to infants born to women infected with zika virus in pregnancy, comprising microcephaly, arthrogryposis, contractures, epilepsy, bone deformities, hearing and ophthalmic alterations, dysphagia, motor and learning difficulties, among others [4,5,6]. A study reported that 45% of the infected fetuses/newborns have no abnormalities compatible with prenatal Zika virus exposure [7], and long-term complications are still under investigation.

In Brazil, between 2015 and 2019 there were approximately 3500 confirmed cases of children born with congenital Zika syndrome (CZS), although the true number may be far higher. Most of these children (63%) live with their families in the country’s northeast region. The state of Pernambuco was the epicentre of the epidemic and recorded 13.5% (471) of the total confirmed cases [8].

Data from the state’s Department of Social Welfare show that almost 90% of these families are registered on the Brazilian Government’s Registry for low income families (CADÚNICO), with a monthly per capita income varying from less than USD 23 (extreme poverty) to the level of the minimum wage (approximately USD 255) [9]. The fact that the poorest families had the highest incidence of Zika infection, and consequently CZS, was the result of factors related to inequality and other social determinants in Brazil, such as low coverage of basic sanitation, intermittent water supply, precarious living conditions and poorly planned urban environments [10,11,12]. It also reflects gaps in access to health services and systems experienced by poorer families [13].

During the Zika epidemic, mothers with babies with CZS were the public face of the epidemic. Mothers reported panic and social guilt knowing that the vertically-transmitted congenital syndrome resulted from them being infected by Zika [14,15]. Women, who in Brazil, as well as elsewhere are usually responsible for assuming the role of carer, took on the major caring role which involved a burden that went beyond the (productive and reproductive) workday [16]. Moreover, mothers of children with CZS assumed an important political role during and immediately after the epidemic in the demand for rights for themselves and their children [17,18]. Public health policies and messaging aimed at controlling the epidemic in Brazil also placed the burden of prevention on women, both in their responsibility for cleaning the home to reduce the spread of mosquitos and in control over reproduction to avoid the birth of affected children [19]. This focus failed to acknowledgement of the lack of reproductive rights and services available for many women in Brazil. 

Living with people who require high levels of support, assistance and care in the family context, such as people with disabilities, the elderly and those with chronic illnesses, can have an impact on and lead to changes in the structure of the lives of family members who take on a more central caring role. Studies focused on family carers of people with disabilities have emphasised that this often involves physical, psychological and financial impacts, changes in perceptions about their own identity, uncertainty about their future, and difficulties with accessing health, education and social welfare services [20]. The literature also notes the capacity for resilience and the benefits arising from the role of family carer [21].

These impacts, when observed through the lens of gender and socio-economic inequalities, disproportionately affect women, in particular mothers. In this article we explore how women gave meaning to the experience of having or caring for a child with CZS. We consider how they believe it impacted their and their families lives, what they have had to let go of, but also what they gained and achieved. In this article we draw on two theoretical approaches to analyse how mothers and other female household members were affected by having a baby with CZS. First, we present the theoretical work that looks at women’s “care work” and how this developed over time, and secondly, the concept of “biographical disruption”. We argue that although these approaches are often used separately, they actually together provide a good way to explain and analysis the way women’s lives were impacted.

### 1.1. Women and Care

Since the second half of the 20th century, feminist historians have examined the concept of “care work”. Key themes researched have included the history of family and how it is constructed, intercultural comparisons on motherhood, fertility, birth, constructions of the notion of childhood and the social hygiene movements. A considerable amount of work has also been produced on the sexual division of labour. A key theme in much of this work has been concepts of domestic work and reproductive labour and this has highlighted how patriarchal relations have excluded women from other forms of social participation over time. Moreover, caring has been commonly removed from the notion of labour and, as with other domestic work, is understood as a less valuable occupation, demeaned and made invisible within families, and in its relationship to economic and social production and reproduction [22].

The idea about an intrinsic assignment of care roles and responsibilities of women varies throughout history and in different cultures, as described in essays by Badinter [23] and Chodorow [24]. Rather than being expressions of a supposedly feminine nature, maternal love and the connection between women and care work are social functions constructed and mediated by hegemonic medical (including, for example, psychoanalysis and psychology), philosophical and political discourses which have varied over time. Since the 18th century, these constructions have influenced and reinforced a model that uses biological functions to justify certain social behaviours, classifying those who diverge from such roles as abnormal or sick. For example, women who abandon motherhood are, even today, frequently considered to be those who have a “defect” in their “real” role as mothers [25]. 

The neglect of the burden of these female roles and the marginalization of care work have been incorporated into the feminist political agenda, which has tried to increase the visibility of these issues to achieve greater recognition and gender equity. The broad agenda regarding women’s autonomy, and the main focus of feminist action, has been in discussions about sexual and reproductive rights, and the possibility of political action and participation in the labour market. The demand for crèches, for example, is an old battle of the feminist movement and reflects the need for public policies that balance the distribution of care work, while allowing greater autonomy for women [26].

### 1.2. Biographical Disruption and Chronic Conditions

The concept of biographical disruption has been used in studies investigating the impact of chronic conditions on how people live their lives and biographical narratives arising from illness. This concept is also useful for analysing the experience of having a child, or being a family carer of a child with disabilities. 

Biographical disruption as a health concept was first used by the British sociologist Michael Bury, in 1982, to refer to the way certain chronic illnesses emerge in an individual’s life at a certain time, modifying their way of being in the world [27]. Chronic illness is thus a disruptive event which mobilizes cognitive and material resources, affecting expectations and plans for the future. For the author, there are three important aspects in the experience of biographical disruption that arise from the appearance of a chronic illness: 

(a) Identifying symptoms and recognizing the fact (disruption of behaviour and assumptions)—the appearance of a chronic illness does not generally happen suddenly, rather it insinuates itself through symptoms that allow the individual to recognize that something is not right and that a modification to their life course, from a trajectory considered normal, is underway.

(b) Uncertainties related to the new life situation (disruption of explanatory systems)—certainty about the disease resulting from the diagnosis gives way to uncertainties about life, since medical and technical knowledge is insufficient and limited in predicting what the impact of living with the disease day-to-day will be. Without certainty about what tomorrow will bring, people use explanatory resources and cognitive organizational structures that go beyond medical explanations and are often also linked to moral concerns, such as their social view of the disease.

(c) A restructuring of emotional, financial and material resources (disruption of the mechanisms to mobilize resources)—the process of becoming ill thus triggers the mobilization of different resources, such as social support networks, access to health services, adaptation of the work environment and the distribution of financial resources, which may lead to huge differences in how the condition manifests in their lives.

Some studies have outlined the limitations of using this concept of biographical disruption. A notable example is the review undertaken by Williams [28], which highlights the inefficiency of the idea of biographical disruption in certain chronic diseases. He argues that health conditions which, although chronic, are expected or considered commonplace, such as diabetes in a family with a family history of the condition, do not cause biographical disruption. Similarly, the same goes for some chronic genetic conditions in children which may be understood as providing continuity, rather than disruption, since they develop very early in the life course.

Williams’ comments are highly pertinent to the broader theme addressed in this article, since it is obviously possible to suppose that, in the case of children affected by CZS, the disease and the complications arising from it do not represent a disruption in their biographies. On the contrary, they not only present biographical continuity, but are a constituent factor of their lives—their mode of being in the world, from birth. On the other hand, the central feature of our analysis focuses on women who have children with CZS, and other family carers. In this way, we highlight how the concept of biographical disruption may be useful for analysing the contexts and life trajectories of women who have children with long-term illnesses or disabilities, rather than focusing on the child with chronic conditions. In this case, although the women are not themselves ill, their children’s disabilities mean that their lives are fundamentally affected. Their behaviour and assumptions have also had to change, in order to address uncertainties and reorganize resources in a similar way to those having to deal with some chronic illnesses. It is worth noting that the term “impact” is used here to describe the result of an unexpected event that leads to modifications or disruption, and may be seen as an improvement to, or a complication of, their previously imagined life course.

In the particular context studied here, disability is an issue arising from a congenital syndrome that affects children during gestation, as a consequence of a pregnant woman becoming infected by the Zika virus. Science had no knowledge or explanations about this condition prior to the birth of the first babies in Pernambuco, the state in which the research was conducted.

## 2. Materials and Methods 

This article forms part of the Social and Economic Impacts of Congenital Zika Syndrome in Brazil Study, whose complete protocol has already been published [29]. The study sample presented here is the result of a qualitative, exploratory and analytic analysis, examining 23 interviews conducted between April and November 2017, with 16 women who gave birth to children with CZS (see Table 1) and 7 other female family carers (4 grandmothers, 1 great-grandmother and 2 aunts). The children with CZS were approximately 1.7 years old at the time of interview. All the women lived in the metropolitan region of Recife, in the state of Pernambuco, northeast Brazil, which was the first epicentre of the 2015–2016 CZS outbreak in Brazil.

Women were eligible for inclusion in the study if they were responsible for the care of a child diagnosed with CZS in their family. They were recruited from other studies underway at the time [30,31] and were then contacted by telephone and asked if they were interested in being interviewed. If they agreed, an interview was arranged at their place of residence or in another location suggested by them. Semi-structured interviews were conducted by trained researchers using an interview guide with themes related to their lives, including pregnancy and reproductive history, information about Zika, health systems and social support. The interviews were carried out in a way that enabled the women to reflect and interpret their life history and to analyse their experiences. The interview guide suggested some thematic approaches, but the women’s responses could influence the interviewer to discuss other themes through reflections and experiences that they chose to talk about. The interviews lasted approximately 40 min each and were recorded using a digital recorder and field notes. All the interviews were transcribed, guaranteeing the participants’ anonymity. All interviewees’ names used in this paper are fictitious.

Following systematic readings, data were organised into an Excel spreadsheet according to large themes agreed in meetings with the study team and based on the interview guide. This was created by at least one person in two research sites from the Social and Economic Impacts of congenital Zika syndrome in Brazil study [29].

For the analysis proposed in this study, a person from the research team was directed to look at the data and focus on particular themes. We were particularly interested in how women explained and interpreted their lives, and whether and how these women’s lives changed following the birth of their child with CZS. After this, an expanded consensus meeting shared the data in this particular context in order to provide feedback, modification, and addition.

Rather than focusing on “facts” we explored descriptions of their life stories and how discourses were constructed around the daily experiences of mothers and carers of children who were born with an unknown disability during an alarming national and international epidemic.

Ethical approval for the full study was received from the London School of Hygiene & Tropical Medicine (LSHTM) and the Oswaldo Cruz Foundation (Fiocruz) ethics committee (Certificate of Presentation of Ethical Appreciation number CAAE 60682516.2.1001.5269) After reading the informed consent form, all participants gave consent; this authorization was made verbally and recorded on audio, and the researcher was responsible for signing the form to register that this had occurred. These precautions were taken in order to ensure the women’s safety, in case information was revealed about abortion or other behaviours considered illegal in Brazil.

## 3. Results and Discussion

For explanation of the results, we have mainly focused on the life stories of three interviewees. They were selected as they allowed a closer and more detailed insight into their biographies, a fundamental point of this paper. We chose these subjects considering the extremities of their circumstances (for the mothers—one who had the best and one who had the worse living condition, and for the carers—the oldest one), allowing us to better visualize and highlight the main points of our results. Although these three narratives contain specific, and even contrasting biographical trajectories, they touch on aspects that can also be observed in the experiences of other participants involved in the research. They therefore serve as indicators for the meanings, values and attitudes related to the impacts that the arrival of a child with disabilities had on the lives of all those we interviewed. A brief life story of these three women can be found in Table 2.

### 3.1. Women’s Life Prior to the Arrival of the New Child

The women interviewed almost all described a usual routine that included attempts to juggle domestic work, childcare (when they were mothers) and sometimes care of elderly relatives, work and study. The women were predominantly from low-income families, and reported lack of money as a constant concern, and with the prevailing patterns of the sexual division of labour, this meant that they did not feel they had choices but instead their lives were limited by pre-existing harsh circumstances.

In this context, finding out that they were pregnant led to a series of changes in their and their families’ lives. When the pregnancy was planned and wanted by the women and her family, these changes were more easily incorporated into daily life. This ideal, however, does not reflect the reality of many of the women interviewed.

“I didn’t imagine [that I was pregnant] because the other one was still a baby and I was taking medication [contraceptive pill]. I was desperate. I said that I was going to give up the child so many times. I was stunned […] like a mad woman, thinking of doing something I shouldn’t.” (Eva)

“At that point I didn’t think I could get pregnant, because I had my first child and I hadn’t used any contraceptive method since [for 10 years]. The doctor said I was ovulating, but I wasn’t fertile and I needed a year’s treatment to get pregnant. So, I wasn’t interested any more. I’d had a dream of having a girl, but I’d already given that up. He just appeared!” (Sara)

Women reported dealing with an unexpected or unwanted pregnancy that interrupted their lives, sometimes leading to feelings of desperation or alternatively to cognitive readjustment that helped them quickly adapt to the new reality. Even for the women interviewed who reported a planned pregnancy, its confirmation leads to a reshaping of their lifestyle. In this sense, the pregnancy itself may be understood as a transformation stage in the life course, raising new challenges and/or feelings of ambivalence in the face of the woman’s own subjective expectations, and given the social roles and moral meanings linked to motherhood [32].

The high number of the women reporting unplanned pregnancies in our study reflects real challenges in terms of the provision of family planning services through public health centres in Brazil, which currently does not adequately deliver information or contraceptive methods needed to support choices related to women’s reproductive intentions. Studies demonstrate that approximately 55% of pregnancies in Brazil are unplanned [33], highlighting the failings in family planning policy and access, which have essentially been centred around guidance about and the prescription of hormonal contraceptive methods [13,33]. This focus ignores the importance of the social context, which, in practice, prevents women from making choices about their sexual and reproductive life, such as poverty and the concurrent lack of access to education and health services and a lack female empowerment necessary for women to make assertive decisions about their lives [34,35,36].

### 3.2. A child with Microcephaly, Now What?

Women reported that whether the pregnancy was planned or wanted they knew that the arrival of a child would impact on family life. However, when this child is born with an obvious severe disability then the impact is essentially different to the impact of having a child without a disability. Women recounted that the moment the disability was diagnosed, and the way in which this diagnosis was given to families had a fundamental impact on how the family understood the medical aspects related to the disability and how they prepared to meet the future challenges related to the child’s complex health and care needs [37,38,39].

Eva and Sara found out about their child’s microcephaly following birth “in the worst possible way”. Some months after adapting to the surprise of being pregnant, the revelation of the potential disability of the newly born child became another moment of disruption in their life trajectories. This impact was aggravated by their health professional’s method of informing them about their child’s condition or in dealing with the anguish of family members.

“they took my baby girl away and they were in there with her for a long time. I thought she’d been born dead. I never imagined she’d been born with microcephaly. A long time afterwards they came and talked to me about her.”(Eva)

“I only found out at the time of the birth, in the worst possible way, when the doctor picked him up and said she needed to take him away because he “wasn’t normal”. So, we only received the diagnosis the following day, which was when she came back to the room.”(Sara)

Some of the women interviewed were given the diagnosis while still pregnant. They reported conversations which they felt were equally upsetting with health workers who were evasive when asked about the implications of the diagnosis.

Albuquerque et al. [13] outline the social and cultural distance between health professionals and healthcare users, in the context of CZS. They highlighted communication difficulties at the point of a diagnosis, where it was made in a unidirectional manner without consideration of the views and position of the family members. Women reported that the way that the diagnosis was made impacted on their future interactions with the healthcare system, and how they accessed other health services the child required. Other papers have explored how the fact that this was a new syndrome, whose characteristics were still poorly understood, also led to health professionals’ insecurity; they often felt ill-equipped to deal with communication with the families about the child’s condition and likely future trajectory [40].

The moment of diagnosis in women’s narratives is clearly another key time of biographical disruption. The diagnosis created a disruption of assumptions related to the idealized image of a child. Women reported it as a moment of recognition that something was happening outside their and their families’ expectations. They realised that they were not going to have the child that she had imagined [27,41].

The alarming national scenario and public health emergency generated insecurity and a rush for responses and care strategies, because of uncertainties about CZS and the children’s prognosis, which also affected relationships within families. The press and the internet became the main source of information about the probable development of their children. However, according to the interviewees, these sources did not reassure them.

“And when I got home I spent all my time searching around on the internet. I “lived” on Google and that was really bad for me because you see the worst possible things. You see that your child will live in a vegetative state, that your child will never walk, that your child will never talk, that he won’t be able to hear. Just the worst things that the internet shows, what it says, right?” (Sara)

This inadequate and limited information triggered the second aspect of biographical disruption, when the formal (medical and technical) explanatory systems failed and were replaced with learning based on experience [27,41]. Technical explanations and trust in the health providers thus gave way to new ways of thinking about the world, and the health system, forged out of daily uncertainties [40]. In the case of Sara, her experience allowed her to challenge the assumption from internet searches that her child would live in a vegetative state:

“While, in fact, daily life was showing me something completely different. My son can see. My son can hear. He can’t talk yet, but he can call his grandma, he can make certain sounds. So, walking is the only thing we’re still waiting for…” (Sara)

These explanatory mechanisms may also trigger transcendental or emotional interpretations of CZS. Eva drew on ideas of guilt and divine retribution when pursuing explanations about her previous life experiences to try to understand why she had a baby with CZS. Twice in her interview she reflected on her current life situation and concluded that it was possible that it was God’s punishment for a previous abortion or the result of prejudiced statements and views that her daughter’s father previously had about people with disabilities.

“when he saw someone like that, in a wheelchair, he was disgusted. At the time, I said to him ”this thing that you do to these people, you turn your back, be careful, because in front of God, he could punish you”. I always said that to him. [...] That’s why I say they’re [father and his family] prejudiced, he doesn’t give my daughter any help. He doesn’t pick her up. He says its agonizing for him. […] after I had the abortion, I suffered a lot. What had I done to a child that hadn’t done anything? I was going to go mad, even after I had the abortion. Then I held on to God, I held on to God, my life improved. I got pregnant again, then I thought: “My God! It was a punishment”. What I did… I took out my… so, you know, I killed my son. God gave me another one.” (Eva)

Frequent social and conventional media stories at the start of the epidemic referred to children with CZS as a “gift from God” or “the angels”, conveying the idea of the purity or sacredness of the affected children. The grace attained with the arrival of the child was considered to come through life lessons and in supporting these children in daily developmental achievements. This belief in divine grace contrasted with the previous media messaging which represented the children as “condemned” to a vegetative life, with extreme limitations or probably early death, and was much easier for the women to deal with.

In all the interviews, it was clear that the arrival of a child with disabilities impacted several aspects of these women’s lives. Because they had children with obvious and severe disabilities, women reported that they felt that they acquired the social position of a person with a disability themselves, in terms of limitations, stigma, social struggle, demand to rights and health services. Moreover, having to look after their children left them little time for other social activities and they became defined to themselves and others as just mothers of children with CZS. Yet their identification as the mothers of the “Zika children”, which is how they were portrayed by the press at the time of the epidemic, created a shared identify, which women reported to lead to a new position of social belonging. The initial solitude arising from the position of being a victim of a “tragedy”, was transformed by linking with others mothers, to a new mission, a God-given opportunity, to show that they were capable “despite everything” of looking after their child. Women reported a strong sense of solidarity they gained from relationships with each other.

### 3.3. Points of Mutation

The interviews show that the women’s lives were dominated by efforts to obtain social benefits and arrange appointments at health centres for consultations with the appropriate health workers for the care of their children. Getting their child the treatment and therapy needed was challenging because of the difficulties of moving around the city and the fragmentation of service provision. This challenge has been documented in a range of studies, such as Albuquerque et al. and Fleisher [13,40,42]. This was, as always, made harder by class and income inequalities, with poorer women of less social capital particularly facing great difficulties in accessing opportunities for supportive therapies for their children’s development. These social inequities may be perceived in Eva and Sara’s different living conditions, which reflect distinct opportunities for autonomy and care.

Eva’s low levels of education and income, and her responsibilities for caring for her four other children made it more difficult for her to access information and health services and receive social benefits. This situation is shared by most of the affected families.

“I stopped [working] when she was born. […] After I began receiving the benefit [The Continuing Benefit Conveyance of BRL 954.00 or USD 243]. Medicine, I buy it when it runs out [...] I receive donated nappies. Sometimes not enough. When that happens, I have to buy them. And at the Social Services Referral Centre (CRAS) I receive a [basic] food basket. […] I chase it up. There was a [Social Worker] who visited us here, at the beginning, when she was a few days’ old. Then they stopped coming. After the benefit was granted, lots of people went away. It isn’t like it was before, when they were on top of everything, they provided that support, I don’t know what. […] This [doctor] now, it takes three months to get a consultation! The only health support going well for me, the only one is at the Foundation [where she has weekly physiotherapy].”(Eva)

For women with similarly scarce resources, the task of obtaining access to services for their children turned into an all-consuming mission. In contrast, those women we interviewed with more resources, had easier access to services.

“Seven o’clock in the morning he has physiotherapy [at a philanthropic service]. From 1.30 p.m. he has occupational therapy covered by private health insurance. That lasts 45 min. At 2.30 p.m. he has physiotherapy covered by private health insurance. That goes until 3 p.m. Then he goes out again for occupational therapy [at the philanthropic service] at 3.50 p.m. and I wait there until 6.10 p.m. for him to do hydrotherapy. [...] He also goes to school on Mondays, Wednesdays, Thursdays and Fridays. On Thursday mornings he has fortnightly visual stimulation [at a philanthropic service]. In the afternoon he has physiotherapy from 4.30–5.30 p.m. On Thursdays he has swimming from 8 to 8.30 a.m. After that he has visual stimulation at 10 a.m. [...] He goes to school on Friday mornings. In the afternoons he has occupational therapy from 1.30 to 2 p.m and I go [to the philanthropic service] at 3.50 p.m., where he has speech therapy. And on Saturdays he has physiotherapy at home, which is private.” (Sara)

This quote illustrates well the different trajectories and biographical pathways that acquire new meanings and new social belongings due to the birth of a child with CZS. These experiences are influenced by the social, political and economic structures in which these women operate. Of all the interviewees, only Sara, who had access to more resources, was able to carry on working following the birth of her child. All the others who had been involved in some kind of work activity had to leave their jobs, either because they decided to or because they were dismissed. This decision was influenced by the difficult reality of having a child with complex health needs. This discontinuation of working further increased the impoverishment in families who then needed to mobilize financial resources for medication, transport, food supplements and other unforeseen demands, although there is a public health system in Brazil. Cepeda et al. [43] also found situations of impoverishment and indebtedness as result of having to pay for medical services in the Dominican Republic.

Developing different ways of mobilizing financial, relational and material resources is the third aspect of these women’s biographical disruptions that emerged from the interviews. [27,41]. When questioned about the changes that had occurred in their lives following the birth of their child, many described the interruption of personal projects, such as work, studies and the desire to have other children, as well as difficulties in carrying out their routine activities and caring for other children in the home. Life impacts arising from having a child with disabilities in the family are not necessarily interpreted as tragedies by these women, whose capacity for resilience and negotiation when confronted with difficult situations is routinely tested.

Authors, such as Shakespeare [44], have used the term *predicament* in reference to disability, to move away from the concept of tragedy and instead focus on the complexity of disability as a difficult situation, which may be painful. Predicament includes the possibility of conquest and escape, unlike a tragedy, which is a negative description. There are social obstacles which intensify these difficulties, but which may be removed in order to minimize them. This approach is useful for reading our study data. At the start of the epidemic, there was a widespread expectation of materialized social tragedy based on the media’s sensationalized images of the non-conforming bodies of hundreds of children with microcephaly. Over time, this account has slowly given way to the protagonist narrative of the women, mothers of these children who have denounced the difficulties and burdens due to the lack of existing alternatives, while pointing out the possibilities that do not end with them. As one of the interviewed mothers says:

“I would like to say that despite all the difficulties we have, this rushing about, our struggle, some people tell the public that we suffer and deserve pity. No! Is it difficult for us? It is. But we are very happy to stand beside our children. I don’t know if I will ever get used to this routine. I wouldn’t change anything. So life is taking me where I need to go.”

This also signals for a notion of care that goes beyond common ideas related to burden activities— and work. In the context of these women’s lives, the idea of care as a vectorial line from who cares, to who is cared for, is contrasted to care as a dimension of intersubjectivity [45,46]. That is, a sharing between women and also between women and their children and between them and health professionals based on the uncertainties and vicissitudes of their daily lives.

Likewise, in contrast to the idea of burden or fatality for women’s lives, there is the ethical dimension of care highlighted by some authors from the concept of interdependence. This concept brought to the foreground that being dependent is an attribute of all people. To some extent, every person at some point in life was, is, or will be dependent. This is, therefore, a social issue that cannot be restricted to the private sphere and cannot fall exclusively on women [47,48]. When care is seen as an individual responsibility, these ethical and social dimensions are lost. Additionally, it represents a failure of the state and its weak governmental policies to take care of its citizens, shifting the responsibility (and blame) to individual actors, with little acknowledgment of structural factors that intersect with infection and contagion.

### 3.4. Biographical Continuities and Support Networks—Who Are the Other Carers?

Support networks are essential for facilitating women in the care of the children and for maintaining the possibility of engaging in personal projects beyond motherhood. These mothers are the children’s most present carers, but other women in the families such as grandmothers, aunts and adolescent sisters are also called to the “mission” of sharing care assignments, either in supporting healthcare needs of the child or by sharing domestic activities. Our data clearly show that these caring roles were taken up by different generations of women in families. Women who understood that there was the possibility of greater autonomy for fellow women if they took on more work and limited their autonomy.

Men rarely appear in these women’s narratives. When present in the children’s daily lives, fathers appear to reproduce socially expected roles and are responsible for sustaining the family or, occasionally “helping” with domestic routines to alleviate the centrality of maternal care [43].

In Sara’s life trajectory, it was her mother-in-law who gave up her house and personal business to live with the couple and her two grandchildren, assisting with the care of her youngest grandchild. It is likely that without this support Sara would not have been able to maintain her over 10-year long career in the same company, while attending her child’s medical appointments, despite the fact that Sara was of a more favourable economic status than most of the affected families.

“My mother-in-law left everything and came to live with us to take care of him. She had a snack bar. She closed it, gave it up and came to live with us. When I went back to work, she came to live with us. She gave us every support, you know. When I can go to the therapy, I go. When I can’t, I leave her with him.”(Sara)

Similarly, Eva’s mother, who for years has shared her house and income with her children and grandchildren took on much larger caring roles. Eva was also helped by her two older daughters, although only 12 and 6 years old.

“And: who looked after them [the children] when I was working? I left him with my oldest girl and the little one. They looked after him. Even today.”(Eva)

These strategies are used by (and between) women trying to fill a gap in state social protection. State participation in this network only appears in the narratives of the interviews as an explicit demand for crèches and schools adapted for children with disabilities.

The life of Maria, the third interviewee whose life trajectory is analysed in detail in this article, is symbolic of someone for whom her grandson’s disability is “just” another of the many circumstances that intersect a biography of frequent disruption. The eldest daughter of a family with many siblings, Maria took on the maternal role following the death of her mother and has worked to support her family since childhood. With her first marriage at 15 years old and her first pregnancy at 16, Maria is mother to 11 children, six of whom live with her and one of whom, now 22 years old, has Down’s syndrome. The arrival of her grandchild with CZS added to the list of life-long difficulties she has faced, in particular, difficulties of access and prejudice regarding her child with disabilities.

“when we discovered that he had microcephaly, my world fell apart, I began to cry because I had this [child with Down’s syndrome] and I know what I suffered when he was small.”(Maria)

Although she recognizes some progress in current public policies compared to her childhood, she remains pessimistic about the future; this appears to be because of the recurrent difficulties she has experienced throughout her life.

“In the past we didn’t have the life we have today, free high school, school meals, uniforms, we have this, that and the other, and in the past we didn’t have any of it […] the future will be even worse than everything we have been through. Life will be even more difficult, from here on it’s going to get even more difficult. […] the doctors even say that these babies disturb the state. […] It is forever, without end, amen!”(Maria)

The interview data suggest that the feelings of repetition and permanence of vulnerabilities across generations of women, many of whom live in poverty and feel abandoned by the state, were intensified within the context of the CZS epidemic. Public policies were formulated without taking account of the time required for the care, and this resulted in the state appropriating the women’s time and reinforcing gender roles that maintain women, almost exclusively, as mothers and carers. Underlying much of the policy approach, was a strategy of making women mainly responsible for family work; to maintain their children, their homes and their own lives [49,50].

It is also worth noting that these women’s social support networks also function outside the family sphere. Social media networks deserve special attention, in particular WhatsApp, which women reported using as a tool for interaction, communication and the exchange of experiences. It was through this tool that one of the main associations for women with children with microcephaly was set up in the state of Pernambuco. This group has become an important collective, powerfully advocating the state to establish the necessary public policies, for both children with CZS and their families, and to provide visibility to the wider struggle of people with disabilities [17,18].

## 4. Conclusions

The births in 2015 and 2016 of children with congenital Zika syndrome in Brazil were unprecedented events that mobilized society and the national and international health authorities. In this context, the symbolic role of mothers gained prominence, either as victims, or protagonists in the struggle for social inclusion and access to meet their children’s complex needs. These women’s life trajectories reveal disruptions and continuities that required them to reshape how they conceptualised their own lives and their social relationships. They made these adjustments in the face of uncertainties, and activated their social support networks, particularly in their intra-family relationships with other women.

We found that the concept of biographical disruption can be useful to analyze and try to understand life experiences from disabled child caregivers, especially when this person is a woman, since they are the ones who—in general—assume the role of carer of the children. Although chronic disease or disability does not cause any deterioration in the capabilities of the body of the caregiver, women caregivers can feel the basic structure of the illness, through disruption of their biography due to tending to the needs of a disabled child.

Revealing the pathways of these biographical narratives allows us to understand how these women have acted to defend the value of their children in a society structured on the model of body normativity, exclusion and inequality. At the same time, it points to the maintenance of vulnerabilities and the imperative need for these women to form their own networks of care and social support in the face of state absence over many generations. Seeking to understand some of the impacts of the Zika epidemic on the women’s lives and how the different contexts of social inequality influence opportunities for them and their children, may provide clues for the pathways of social transformations that are required for a more inclusive society. These transformations must confront systems of gender oppression and the sexual division of labour in social relationships, without giving up the role of the state.

## Figures and Tables

**Table 1 viruses-12-01410-t001:** Characteristics of women interviewees who had children with congenital Zika syndrome (CZS).

Age (in Years)	Race/Colour **	How Pregnancy Occurred	Age of Child	When Diagnosis Occurred	Age of Other Children	Work Situation		Marital Situation
						Prior to Birth	After Birth	
20	Mixed Race	Beginning to think about it	1 year, 8 months	Immediately after birth	Single child	Never worked	Never worked	Married/living with partner
36	Mixed Race	Planned	1 year,78 months	During pregnancy—5th month	Single child	Cashier	Stopped working	Married/living with partner
18	White	Unplanned	1 year, 8 months	During pregnancy—7th month	3 months	Was not working	Was not working	In a relationship
42	Mixed Race	Adopted	1 year, 8 months	Immediately after birth	2 adults—age not given	Domestic work	Stopped working	Widow
* 34	Not informed	Unplanned	1 year, 3 months	Immediately after birth	12, 6, 4 and 2 years old	Domestic work	Stopped working	Single
Not informed	Black	Unplanned	1 year, 10 months	Immediately after birth	9 years old	Lift operator	Stopped working	Married/living with partner
31	Not informed	Unplanned	1 year, 10 months	During pregnancy—7th month	Single child	Worked in a shop	Stopped working	Married/living with partner
* 39	White	Unplanned	1 year, 9 months	Immediately after birth	19 years old	Banker	Banker	Married/living with partner
27	Not informed	Planned	1 year, 6 months	Immediately after birth	8 years old	Was not working	Was not working	Married/living with partner
Not informed	White	Planned	1 year, 10 months	Immediately after birth	18 years old, 16 years old, 4 months	Worked as a daily cleaner	Stopped working	In a relationship
23	Mixed Race	Planned	1 year, 8 months	During pregnancy—7th month	Single child	Never worked	Never worked	Married/living with partner
35	Mixed Race	Planned	1 year, 7 months	Immediately after birth	Single child	Worked (did not say where)	Stopped working	Married/living with partner
23	Not informed	Planned	1 year, 10 months	Immediately after birth	7 years old, 4 months	Domestic worker	Stopped working	Married/living with partner
28	Not informed	Planned	2 years	During pregnancy—7th month	Single child	Domestic worker	Stopped working	Married/living with partner
35	Mixed Race	Planned	1 year,11 months	Immediately after birth	Single child	Administrative Assistant	Stopped working	Separated
33	White	Unplanned	1 year, 8 months	After birth—5th month	14 years old	Selling cosmetics—freelance	Stopped working	Married/living with partner

Created by the authors. * participants marked are those who we selected to illustrate our findings in this paper. ** in Brazil the categories “colour” and/or “race” are commonly used with the same meaning to denote racial patterns and identification with racial group memberships.

**Table 2 viruses-12-01410-t002:** A brief life story of three women interviewed.

Eva * is 34 years old and has five children. She was single at the time of the interview; her last relationship lasted six years and broke down as a result of recurring acts of violence and her partner’s involvement in drug trafficking. Her ex-partner is the father of Eva’s three youngest children, one of whom, aged four, lives with his father but is cared for by his paternal grandmother. All Eva’s other children—including her daughter *Carla **, who is 1 year, 3 months old and was born with CZS/microcephaly—live with her in a small two-room space attached to her mother’s house provided by her mother so she could live with her children. Prior to Carla’s birth, Eva used to work during the week as a “clandestine” domestic worker, receiving less than half a minimum wage per month; she stopped working when her daughter was born. Her pregnancy came as a surprise and made her feel “desperate”, because she already had four children, one of who was only a few months old. The news about her daughter’s disability came after her birth. Pregnancy and her daughter’s birth led to certain changes: she left the abusive relationship, she left her “clandestine” job, she stopped going out to have fun, she began to receive a government benefit of one minimum wage and she had difficulties finding services adequate to meet the needs of her daughter, who began occupational therapy and physiotherapy stimulation at 8 months, once a week, at a clinic linked to the public sector in a hard-to-access location, which relies on public transport. Her mother and 12-year-old daughter help to take care of the children.
Sara * is 39 years old, married with two children, one 19 years old and Julio ***** who is 1 year, 9 months and was born with microcephaly/congenital Zika syndrome. She was trained in business administration and had a 10-year career in the private sector. The child was not planned. The pregnancy came as a surprise, after using the intrauterine device for 9 years and following a 10-year period without using any form of contraception. The news about her child’s potential disability came immediately after birth, and was later confirmed through tests and consultations with specialists, who informed her of the need to initiate early stimulation immediately. The arrival of her son led to a new movement in her life which, from then on, had to reconcile the demands of work, a range of therapies for her son, activities at the mothers’ association and conjugal, family and domestic life. Since he was two months old, her youngest child’s routine has included repeated and varied therapies. At the time of the interview, Julio was undergoing occupational therapy, speech therapy, hydrotherapy, visual simulation, swimming and school activities throughout the week. Some of these activities are provided by the public sector and others are paid for by the family, either directly, or through their private health insurance. Her mother-in-law went to live with the family to support her grandson’s care routines.
Maria * is 63 years old, grandmother of Laura who is 1 year, 9 months old and was born with microcephaly/congenital Zika syndrome. She studied until the fourth year of primary school and for the best part of her life has worked as a domestic worker; she began this work aged 12. She has lived with her second partner, the father of 7 of her 11 children, for more than 30 years. Her first daughter died in a meningitis outbreak; the others are alive. Six of them live with her, including Laura’s mother who returned to her mother’s house when she was six months’ pregnant following the breakdown of her relationship with the partner she was living with. Maria also has a son with Down’s syndrome, now 22 years old. Another of her daughters had just lost a baby in the 8th month of pregnancy due to foetal death caused by congenital syphilis. The news about Laura’s disability came after 8 months of pregnancy and initially caused a commotion in the family which already had experience of looking after someone with a disability. Since childhood, Maria has been the person who supports the family, as the oldest daughter helping to care for dozens of siblings.

Created by the authors. * All names used are fictitious.

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
