# Peer review of "“Life Is Taking Me Where I Need to Go”: Biographical Disruption and New Arrangements in the Lives of Female Family Carers of Children with Congenital Zika Syndrome in Pernambuco, Brazil"

_viruses, 2020, doi:10.3390/v12121410_

Round 1
Reviewer 1 Report
Thank you for submitting this manuscript. It is well-written and deals with a ver important topic, foregrounding the important intersections between care, gender, and structural disadvantage. I only have minor comments:
Abstract
- 'social weight from expected maternity and care roles': rephrase for clarity
- sentence starting ‘through the concept of biographical’, would be better positioned after the sentence introducing the study methodology
- last sentence: it is not clear what ‘it’ refers to
Table 1: you use ‘colour’ to denote ethnicity. I suggest rewording.
Table 2 is part of the findings, not the methods
The points below are suggestions, in case you think they are relevant to your rgument:
The background section on care is informative and interesting. You might want to problematise the common conceptualisation of care as burden and discuss the intersubjectivity of care, which is what your findings also highlight.
From the findings and the discussion I get a strong sense that these women had to form their own networks to take care of their children and each other; to some extent, this suggests a failure of the state to take care of its citizens, shifting the responsibility (and blame) to individual actors, with little acknowledgment of structural factors that intersect with infection and contagion. You might want to highlight this.
I look forward to seeing this manuscript in print.
Reviewer 2 Report
This qualitative study, using semi-structured interviews, aims to describe the social arrangements and life disruptions of mothers caring for infants affected by Congenital Zika Syndrome.
Introduction:
The references in the introduction are up to date, but I will add a paragraph on Zika's congenital infection: rate of maternal-fetal transmission, rate of symptomatic infections at birth, rate of later complications.
In the part "Biographical Disruption and Chronic Conditions" of the introduction, it seems important to ackowledge that although CZS is an evolving condition, the lack of its prenatal diagnosis may represent a disruption for mothers at birth.
Methods:
Table1: Does "Sigle child" mean "single child"?
This study focus on three interviews (reported in Table 2). Why only 3 interviews were summarized? How they were selected (objective criteria / Subjectively)?
It could be interesting to present the other interviews in a supplementary Table.
How many co-authors reviewed the interviews and identified social and biographical disruptions in the data? It should be specified in the methods.
Results and Discussion:
-Please, give the proportions when you write "almost all" or "predominaly" or "high number", it is paramount to understand the background of these interviews.
-The transcription of the interviews is well written and easy to follow.
- I would have liked to have a little more information on late anomalies related to CZS spectrum. How these mothers and grandmothers reacted to neurodevelopmental delays dignosticated away from an asymptomatic infection at birth? How did they withstand the heaviest forms with cerebral palsy and pharmaco-resistive seizures?
Minor comments
Please review the syntax of this manuscript:
- L43: Children born with CZS
- L69: This focus failet to acknowledge the lack
- L86: provide a good way to explain and analyze
- etc
